Divergence by depth in an oceanic fish

Shum Peter 1
Pampoulie Christophe 2
Sacchi Carlotta 3
Mariani Stefano 1 s.mariani@salford.ac.uk
1 School of Environment & Life Sciences, University of Salford , Manchester , UK
2 Marine Research Institute , Reykjavík , Iceland
3 School of Biology & Environmental Science, University College Dublin , Dublin , Ireland
Toonen Robert
Electronic publication date: 2014 Aug 14
Publication date: 2014
Volume: 2
Electronic Location ID: e525
Received 2014 Jan 14; Accepted 2014 Jul 25
Copyright: © 2014 Shum et al.
Copyright year: 2014
Copyright holder: Shum et al.
License: This is an open access article distributed under the terms of the Creative Commons Attribution License, which permits unrestricted use, distribution, and reproduction in any medium, provided the original author and source are credited.
License URL: https://creativecommons.org/licenses/by/3.0/

Keywords: Sebastes, Rhodopsin, Environmental gradient, Adaptation, Vision, Deep sea

Funding: University College Dublin The Icelandic Marine Research Institute The University of Salford Financial support was provided by the University College Dublin, the Icelandic Marine Research Institute and the University of Salford. The funders had no role in study design, data collection and analysis, decision to publish, or preparation of the manuscript.

==============================
Despite the striking physical and environmental gradients associated with depth variation in the oceans, relatively little is known about their impact on population diversification, adaptation and speciation. Changes in light associated with increasing depth are likely to alter the visual environment of organisms, and adaptive changes in visual systems may be expected. The pelagic beaked redfish, Sebastes mentella, exhibits depth-associated patterns of substructure in the central North Atlantic, with a widely distributed shallow-pelagic population inhabiting waters between 250 and 550 m depth and a deep-pelagic population dwelling between 550 and 800 m. Here we performed a molecular genetic investigation of samples from fish collected from ‘shallow’ and ‘deep’ populations, using the mitochondrial control region and the gene coding for the visual-pigment rhodopsin. We identify patterns suggestive of potential adaptation to different depths, by detecting a specific amino acid replacement at the rhodopsin gene. Mitochondrial DNA results reflect a scenario of long-term demographic independence between the two S. mentella groups, and raise the possibility that these ‘stocks’ may in fact be two incipient species.

Introduction

Speciation phenomena in taxa diverging with gene flow, in the absence of obvious geographic barriers, remain a central focus in evolutionary biology (Bird et al., 2012; Smadja & Butlin, 2011; Fitzpatrick, Fordyce & Gavrilets, 2008). The classical model of allopatric speciation involves the evolution of reproductive isolation as a result of physical barriers that block gene flow in spatially separated populations; whereas populations diverging in sympatry lead to the formation of species from a single panmictic population which must exhibit strong divergent selection in order to overcome the homogenizing effects of gene flow (Gavrilets, 2003). Parapatric speciation represents an intermediate scenario of species formation whereby partial yet restricted contact zones exist between two populations with limited gene exchange (Gavrilets, 2003). Although exhaustive frameworks exist to identify and interpret speciation dynamics (Rettelbach et al., 2013; Bird et al., 2012; Smadja & Butlin, 2011), only recently has empirical attention been directed to the role of depth gradients in aquatic (Vonlanthen et al., 2009) and especially oceanic biota (Roy, Hurlbut & Ruzzante, 2012; Bird et al., 2011; Brokovich et al., 2010; Hyde et al., 2008).

The strong physical gradients across depth layers in the ocean pose strong selective pressures in aquatic organisms (Ingram, 2011; Somero, 1992), and one notable factor is the change in the light environment (Warrant & Locket, 2004), which affects vision. Visual sensitivity in marine vertebrates depends on the spectral tuning mechanism of the visual pigment (VP) (Yokoyama, 2000), which consists of an opsin protein (part of the largest family of G-protein-coupled receptors) bound to a light-sensitive chromophore. Differently-charged amino acid (AA) residues in the opsin will result in slightly different light absorbance by the photoreceptor cells (Yokoyama, 2002).

The percomorph marine family of rockfishes (Sebastidae) have played a central role in the understanding of depth-associated population divergence and speciation in the ocean (Ingram, 2011; Stefánsson et al., 2009a; Hyde et al., 2008; Alesandrini & Bernardi, 1999), and evidence exists that the rhodopsin gene may have evolved in response to different light environments in the main, ancient radiation of the genus, in the Pacific (Sivasundar & Palumbi, 2010). North-Atlantic Sebastes have a much more recent history (Hyde & Vetter, 2007), with the four recognised extant species having diversified during the Pleistocene (Bunke, Hanel & Trautner, 2012). In particular, the beaked redfish, Sebastes mentella, consists of two genetically distinguishable groups (Stefánsson et al., 2009a; Pampoulie & Daníelsdóttir, 2008): a widely-distributed shallow-pelagic (SP) form, found between 250 and 550 m depth and a more circumscribed deep-pelagic (DP) component, between 550 and 800 m. However, doubts remain as to the forces at play and the time scales associated with this divergence (Stefánsson et al., 2009b; Cadrin et al., 2010).

Here, we sought to investigate whether on-going processes of adaptation to different depth layers may leave a signature of disruptive selection in the rhodopsin gene in a recently diversifying Sebastes species. We also employed for the first time the mitochondrial DNA control region to reconstruct historical demography and to further elucidate the evolutionary relationships between ‘shallow’ and ‘deep’ pelagic beaked redfish.

Material and Methods

Generation of molecular data

Archive samples were randomly selected from 25 shallow-pelagic (SP) (collected above 400 m depth) and 25 deep-pelagic (DP) (collected below 700 m) Sebastes mentella from the Irminger Sea, south-west of Iceland previously genotyped by Stefánsson et al. (2009b) (sample numbers 4 & 5 in the original article). DNA was isolated from gill tissue that had been preserved in 96% EtOH using a modified salt extraction protocol (Miller, Dykes & Polesky, 1988) or the DNeasy kit (Qiagen©) following the manufacturer’s protocol. The non-coding mitochondrial control region was amplified by PCR using primers developed by Hyde & Vetter (2007); D-RF: 5′-CCT GAA AAT AGG AAC CAA ATG CCA G-3′ and Thr-RF: 5′-GAG GAY AAA GCA CTT GAA TGA GC-3′. The primers by Chen, Bonillo & Lecointre (2003); Rh193: 5′-CNT ATG AAT AYC CTC AGT ACT ACC-3′ and Rh1039r: 5′-TGC TTG TTC ATG CAG ATG TAG A-3′ were used to amplify 744 bp of the intron-free rhodopsin gene in 10 shallow-pelagic S. mentella, 12 deep-pelagic S. mentella, and 3 and 4 individuals of S. marinus and S. viviparus as outgroups. Reactions were carried out in 25 µl volumes containing 1× PCR buffer, 1 mM MgC12, 200 µM dNTPs, 0.4 µM of each primer, 0.2 units Taq DNA polymerase, with 1 µl of DNA template for mtDNA (4 µl for rhodopsin). Amplifications were performed in a Biometra T3000 Thermocycler using the following temperature profiles: control region: 94 °C (2 min), 35 cycles of [94 °C (30 s), 59 °C (60 s), 72 °C (60 s)], followed by 3 min at 72 °C; for rhodopsin: 95 °C (5 min), 37 cycles of [94 °C (20 s), 58 °C (30 s), 72 °C (45 s)], followed by 5 min at 72 °C. A negative control was included in all reactions. PCR products were subjected to electrophoresis through a 1% agarose gel 1X Tris–Borate–EDTA Buffer, stained with SYBR green for visualisation via a UV-transilluminator and then purified through the addition of exonuclease I and shrimp alkaline phosphatase to remove unincorporated primers and deoxynucleotides in preparation to sequencing. Purified products were sequenced by Macrogen (Macrogen, Amsterdam; http://dna.macrogen.com/eng/).

Data Analysis

Genetic diversity and population differentiation

The mtDNA control region was examined for nucleotide and haplotype diversity. This included the number of net nucleotide substitutions per site between populations (Da) which was calculated using DnaSP v5.10 (Librado & Rozas, 2009). We estimated the level of genetic variation between populations calculating pairwise population FST and ΦST values performed in ARLEQUIN v3.5.1.2 (Excoffier & Lischer, 2010) with significant values tested by 5,000 permutations.

Mismatch analysis was performed to examine the demographic history between the shallow-pelagic and deep-pelagic S. mentella populations using ARLEQUIN, and distributions were compared with a two-sample Kolmogorov–Smirnov (K–S) test. For populations at stationary demographic equilibrium, theoretical and empirical studies show that the mismatch distributions usually have multimodal, ragged or erratic distributions, while these are typically smoother or unimodal for populations that have undergone a recent expansion (Rogers & Harpending, 1992). To test the goodness-of-fit of distributions, we calculated the sum of squared deviations (SSD) and raggedness index (r) for a stepwise expansion model for the data tested by Monte Carlo Markov Chain simulations (1,000 steps) in ARLEQUIN.

Haplotype genealogies for the S. mentella data set were constructed following a method described by Salzburger, Ewing & von Haeseler (2011) based on a maximum likelihood tree for mtDNA and rhodopsin genes sequences.

Data from a selected suite of 12 microsatellite loci previously used for genotyping by Stefánsson et al. (2009b) were used to calculate pairwise genetic differentiation (Weir & Cockerham’s FST, Hedricks GST′ & Jost’s Dest) between populations with 9,999 permutations carried out to obtain significance levels using GenAIEx 6.501 software (Peakall & Smouse, 2006). Population structure was visualized by correspondence analysis (CA) using GENETIX 4.05 (Belkhir et al., 1996).

Phylogenetic analysis and test for positive selection

Maximum-likelihood (ML) analyses of the rhodopsin gene sequences were performed using PhyML 3.0 (Guindon et al., 2010) under 1,000 replications; using Modeltest3.7 (Posada & Crandall, 1998), the model that best fit the data was found to be F81+ I (pinvar = 0.9770). Trees generated from these results were used for a test for positive selection at the rhodopsin gene, conducted using the Creevey–McInerney method (Creevey & McInerney, 2002) implemented in CRANN (Creevey & McInerney, 2003). This test is a more sensitive tree-based analysis derived from the relative ratio test McDonald & Kreitman (1991). Given an appropriate rooted tree, the number of synonymous and non-synonymous substitutions are calculated along each internal branch using the reconstructed ancestral sequences. The method uses statistical tests for independence (χ2 G-test or Fisher’s exact test) to evaluate whether the ratio between synonymous (Silent Invariable (SI) to Silent Variable (SV)) and non-synonymous (Replacement Invariable (RI) to Replacement Variable (RV)) substitutions deviate from the expected value under the neutral model. Where the G-test fails to produce a result, the Fisher’s test is used and vice versa. Positive directional selection is expected if there is a significantly higher number of RI substitutions or non-directional selection if RV >RI. The test was performed using S. alutus as outgroup (Fig. 3) as it represents the closest common ancestor for the Atlantic Sebastes spp. (Hyde & Vetter, 2007).

Results and Discussion

We recovered 16 mtDNA haplotypes, defined by 15 total variable sites across the two groups. Haplotypes were almost completely segregated between depth layers, with eight haplotypes found in the deep, never recovered in the shallow area, and resembling a starburst pattern, three mutational steps away from the rest of the network (Fig. 1A). Of the 10 haplotypes found in the shallow, eight were exclusive of this habitat, and only two individuals collected in the deep were found to bear a ‘shallow-type’ sequence. The shallow-pelagic (SP) group exhibited much greater diversity (hˆ = 0.887 ± 0.033, π = 0.00504 ± 0.00082) than the deep-pelagic (DP) group (hˆ = 0.543 ± 0.119, π = 0.00238 ± 0.00071) (Table 1).

Figure 1 Comprehensive image of mtDNA and Rhodopsin genetic divergence and Mismatch Distributions.

S. mentella genealogies for mtDNA (n = 50; 25SP + 25DP) and rhodopsin (n = 22; 10SP + 12DP), and mtDNA mismatch distributions. (A) Haplotype network for the shallow (red) and deep (blue) groups for mtDNA (i) and rhodopsin (ii). The size of each circle represents the proportion of haplotypes. The lengths of the connecting lines reflect the number of mutations between haplotypes. (B) Mismatch distributions from the mtDNA sequences of shallow (i) and deep (ii) groups, respectively from above and below 550 metres depth respectively. Dotted lines (Up/low bound.) represent the 95% boundaries of the simulated distributions.

Table 1 Summary of mtDNA control region molecular diversity.

Population	H/n	S	hˆ ± SD	π ± SD	DT	FS	F∗	D∗	
(SP)	8/25	10	0.887 ± 0.033	0.00504 ± 0.00082	−1.08590	−3.806	−2.30037	−2.35314	
(DP)	6/25	8	0.543 ± 0.119	0.00238 ± 0.00071	−1.77639	−4.717	−1.49744	−1.08199	
Notes.

H unique haplotypes

n number of individuals

S Segregating sites

hˆ haplotype diversity

π nucleotide diversity (both with associated standard deviations, SD)

DT Tajima’s D

FS Fu’s

FS statistic

F∗ Fu and Li’s F test

D∗ Fu and Li’s D test

SP Shallow Pelagic

DP Deep Pelagic

Partitioning of genetic variance between the populations showed highly significant and strong population structure (Table 2). The mismatch distributions of the two groups (Fig. 1B) were significantly different (K–S test: D300 = 0.3, p ≪ 0.001), and confirmed what is visually apparent from the haplotype network: a scenario of more recent and pronounced demographic and spatial expansion in the deep-pelagic group compared to the shallow-pelagic (Table 3). Using Nei’s (Nei, 1987) formula for divergence time: T = Da/2 µ, where 2µ represents a general mtDNA evolutionary rate, commonly assumed to be around 11% per million years for fish mtDNA control region (Patarnello, Volckaert & Castilho, 2007), we find that the “deep” and “shallow” lineages split over 44,000 years ago.

Table 2 Analysis of fixation/differentiation indices for mtDNA and microsatellite data between shallow-pelagic (SP) and deep-pelagic (DP) S mentella.

Marker	Group	Fixation/differentiation
index	Estimate	p	
mtDNA	SP vs. DP	FST	0.636	<0.001	
		ΦST	0.273	<0.001	
Microsatellites	SP vs. DP	FST	0.031	0.001	
		GST′	0.135	0.001	
		Dest	0.121	0.001	

The rhodopsin gene was tested for signatures of positive selection using the Creevey-McInerney method rooting the tree with S. alutus as an outgroup (Fig. 2). Values for the four substitution variables, G-test, and p-values along each branch are presented in Table 4. Two branches (numbers 26 and 27) showed significance at the p = 0.05 level (Fig. 2). Branches 26 and 27 show significant RI to RV deviations from neutrality, due to non-synonymous substitutions (Table 3), suggesting that positive disruptive selection is acting on the rhodopsin gene for the clade and on the internal branch leading to the shallow group. We observed a fixed non-synonymous AA substitution within the transmembrane domain, which strongly discriminates the two groups inhabiting shallow and deep environments (Fig. 3). The shallow-pelagic group exhibits a GTC at position 119, which codes for Valine (L119V), while the deep-pelagic type displays an ATC, coding for Isoleucine (L119I). Amino acid changes located in the transmembrane helical regions have been known to be important for spectral tuning (Yokoyama et al., 2008), yet in-vitro experimental spectral analyses of vertebrate rhodopsins suggest that amino acid substitutions at site 119 have negligible effect on absorption spectra (Yokoyama et al., 2008). Nevertheless, substitutions within the transmembrane protein domain III (helix-III), such as site 119, have been shown to affect the decay rate of metarhodopsin II (“meta II”; Ou et al., 2011), which is an intermediate of rhodopsin that binds and activates transducin, the visual G-protein (Smith, 2010). Ou et al. (2011) have shown that an AA replacement L119C against the wild-type rhodopsin resulted in shorter meta II lifetimes, suggesting more responsive structural alterations at the helix’ G-protein binding site.

Figure 2 CRANN test tree.

Creevey–McInerney analysis of Sebastes rhodopsin. Rhodopsin reveals significant positive selection (*) at two nodes (26, 27).

Table 3 Mismatch distribution parameter estimates for mtDNA control region.

Population	Mismatch distribution	
	τ	θ 0	θ 1	SSD p-value ± SD	r p-value ± SD	
(SP)	1.9	0.0000	99,999	DE 0.007 ± 0.18	0.30 ± 0.21	
				SE 0.007 ± 0.002	0.27 ± 0.28	
(DP)	0.1	0.0000	99,999	DE 0.274 ± 0.18	0.00 ± 0.21	
				SE 0.004 ± 0.002	0.67 ± 0.28	
Notes.

τ tau

θ0 theta 0

θ1 theta 1

SSD sum of squared deviations

r raggedness statistic

DE demographic expansion

SE spatial expansion

(SP) Shallow pelagic

(DP) Deep-pelagic

Table 4 Creevey–McInerney positive selection analysis on Sebastes rhodopsin sequences outgrouped with S. alutus (GenBank: EF212407.1), G-value p < 0.05 for Fisher’s† and G-Test*.

Branch no.	RI	RV	SI	SV	G-value	
Rhodopsin						
0	0	2	0	0	0.00	
1	0	2	0	0	0.00	
2	0	2	0	0	0.00	
3	0	5	0	0	0.00	
4	0	5	0	0	0.00	
5	0	5	0	0	0.00	
6	0	5	0	0	0.00	
7	0	5	0	0	0.00	
8	0	5	0	0	0.00	
9	0	5	0	0	0.00	
10	0	5	0	0	0.00	
11	0	8	0	0	0.00	
12	0	8	0	0	0.00	
13	0	8	0	0	0.00	
14	0	8	0	0	0.00	
15	0	8	0	0	0.00	
16	0	11	1	0	2.01	
17	0	0	1	0	0.00	
18	0	9	0	0	0.00	
19	0	11	0	0	0.00	
20	0	12	0	0	0.00	
21	0	12	0	0	2.2	
22	0	12	1	0	2.2	
23	0	12	1	0	2.2	
24	0	12	1	0	2.2	
25	0	12	1	0	2.2	
26	1	12	2	0	5.5†	
27	8	24	5	1	6.76*	

Neither spectral nor conformational analyses have so far been conducted on shallow-pelagic and deep-pelagic Sebastes mentella, but the AA variation observed here could underlie differential hydrophobic activity and photoisomerization sensitivity (Ou et al., 2011) that could hold some adaptive value, even without net change in wavelength of maximal absorption.

Sivasundar & Palumbi (2010) discovered a striking association between AA replacements along the rhodopsin gene and inferred depth preference in many North Pacific Sebastes. Interestingly, four North-Pacific Sebastes (S. chlorostictus, S. elongatus, S. aurora and S. melanostomus) typically associated with deeper waters were observed to exhibit the same AA replacement L119I as detected in the deep-sea S. mentella. Similarly, one Pacific species (S. diploproa) exhibits the AA replacement L119V, which is linked to a shift back to shallower waters, and mirrors the polymorphism in the shallow-pelagic S. mentella.

Although larger sample sizes will be required in the future to test these patterns more robustly, the implications of these findings are twofold, and have powerful resonance for both marine evolution and fisheries management. First, mitochondrial variation between ‘shallow’ and ‘deep’ S. mentella in the North Atlantic unveil a degree of historical divergence that previously employed genetic markers either failed to detect (Bunke, Hanel & Trautner, 2012) or could not reliably frame in a phylogeographic context (Stefánsson et al., 2009b).

The level of differentiation and haplotype sorting is such that evolutionary independence can be broadly upheld for these two habitat-segregated lineages, and re-analysis of microsatellite data confirm this picture (Fig. 4). In particular, the comparison of frequency-based indicators of substructure (FST for mtDNA, and GST′ and Dest to account for the hypervariability of microsatellite loci) reveal values (Table 2) that match theoretical expectations under neutral divergence, taking into account the four-fold strength of genetic drift at mitochondrial markers. Interestingly, we also noticed two individuals with a ‘shallow-type’ mtDNA haplotype, which were caught in the deep layer. One of these two (DP29, Fig. 2) also screened at the rhodopsin locus, exhibits a sequence typical of the shallow layer, and its multilocus microsatellite genotype also falls with the shallow group (Fig. 4), which can be interpreted as the occurrence of individual movements along the water column during the life cycle (i.e., short-term “dives” into the deep, by shallow-dwelling fish). Another deep-caught individual (DP1) also exhibits a “shallow” haplotype, but a “deep-like” ATC rhodopsin sequence and an inconclusive multilocus microsatellite genotype (Fig. 4). Collectively, this likely reflects the occurrence of introgressive hybridisation between the two groups, as previously suggested by Pampoulie & Daníelsdóttir (2008).

Figure 3 Example of non-synonymous base substitution.

Chromatograms illustrating the non-synonymous A/G mutation on the rhodopsin gene, which discriminates between “Deep-Pelagic” (A) and “Shallow-Pelagic” (B) Sebastes mentella.

Figure 4 Ordination of microsatellite genotypes.

Correspondence analysis based on microsatellite data. Each circle represents an individual; red and blue refer to the shallow-pelagic (SP) and deep-pelagic (DP) groups respectively.

Furthermore, the stark pattern of depth-associated divergence at the rhodopsin gene is perhaps even more surprising, were it not for the fact that comparable evolutionary genetic patterns have recently been credited with a key role in the diversification of the more ancient Pacific Sebastes group (Sivasundar & Palumbi, 2010). It has been hypothesized that fast-evolving markers will allow to determine recent speciation events for closely related Sebastes spp. (Alesandrini & Bernardi, 1999; Cadrin et al., 2010). The present data provide a snapshot of the evolutionary mechanisms that may be at play in the young, species-poor, Atlantic Sebastes lineage, during its initial phase of adaptive radiation, underpinned by positive selection at the rhodopsin gene.

Less than a decade ago, S. mentella was assumed to be panmictic in the North Atlantic, and the rapidly increasing fishery pressure on these stocks did not recognise any possible substructure until 2009 (Cadrin et al., 2010). These latest results dismiss the notion of panmixia in this oceanic species, and, perhaps more intriguingly, open the possibility that the two ‘shallow’ and ‘deep’ groups may represent two lineages experiencing adaptation towards divergent environmental conditions. In the near future, it should be experimentally evaluated whether the amino acid replacements at the 119 position actually produce detectable changes in retinal absorbance or structural responsiveness, and whether more powerful molecular comparisons covering a wider portion of the genome (e.g. SNP-based genome scans; transcriptomic approaches) will offer further insights into the role of depth as a diversifying agent in the ocean.

We are grateful to the WKREDS workshop of the International Council for the Exploration of the Sea (ICES) for inspiring this work. We are also indebted to Valerie Chosson for technical assistance at the MRI, and Emma Teeling, Bruno Fonseca Simões and three reviewers, for the constructive criticism offered.

Additional Information and Declarations

Competing Interests

Author Contributions

Animal Ethics

DNA Deposition

The authors declare there are no competing interests.

Peter Shum performed the experiments, analyzed the data, wrote the paper, prepared figures and/or tables, reviewed drafts of the paper.

Christophe Pampoulie conceived and designed the experiments, contributed reagents/materials/analysis tools, reviewed drafts of the paper.

Carlotta Sacchi performed the experiments, reviewed drafts of the paper.

Stefano Mariani conceived and designed the experiments, analyzed the data, contributed reagents/materials/analysis tools, wrote the paper, reviewed drafts of the paper.

The following information was supplied relating to ethical approvals (i.e., approving body and any reference numbers):

UCD Ethics Committee, AREC-P-1023 Mariani.

The following information was supplied regarding the deposition of DNA sequences:

GenBank Accession numbers for the mtDNA (deep and shallow) and Rhodopsin (Sebastes mentella, S. marinus, S. viviparus) sequences: KM013849–KM013927.

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
