# Peer review of "Divergence by depth in an oceanic fish"

_PeerJ, doi:10.7717/peerj.525_

## Round 0.1 · original submission · Major Revisions

· Academic Editor

Major Revisions

Overall all 3 reviewers agreed that your subject was compelling and should ultimately be published, but that there were issues that need to be dealt with before acceptance. While seeing the worth of the subject and the data, all 3 referees also agreed that the conclusions were overly strong for the data. I am of the opinion that authors should be able to hypothesize about what they feel their results show, but also agree with the referees that drawing strong conclusions in the absence of sufficient evidence is premature. I believe that you should easily be able to revise the manuscript as suggested by the referees, and still have a section in the discussion about what you think this finding means for incipient speciation. However I also find myself agreeing with referees that the paper would be better focused on the divergence between depth with a conclusion about implications for speciation rather than leading the entire paper with it.
I likewise agree that this seems like a case in which some additional analyses with IMa2 would contribute to the manuscript,
but I leave that decision to the authors as to whether or not they want to include that information in this manuscript. Additionally as one of the referees points out, you need to clarify exactly what method you used to calculate your measures of genetic differentiation, and in fact it would be good to include a couple different measures and be very specific about which measures you have used as outlined in Bird et al. (2011) - http://www.researchgate.net/publication/229089010_Detecting_and_measuring_genetic_differentiation/file/9c96051835c4fc003c.pdf
Again, this is a suggestion for improvement of the manuscript rather than an editorial mandate.
Overall, I believe that the referees have provided clear and specific advise on revision that will improve the manuscript. If you have any questions about it, please contact me.

Reviewer 1 ·

Basic reporting

Overall, the subject material tackled in this simple experiment are interesting – how and why are marine populations structured by depth? Despite there only being two samples and 50 individuals in this study, I believe it is possible for the data presented in this manuscript to be a worthy contribution to the literature on selectively driven population structure. However, I find the treatment here to be cursory and more akin to a first draft than a polished product worthy of publication in a peer-reviewed journal. Basically, what is presented is the skeleton of a manuscript and it suffers because of that.
The abstract is poorly written. The intro is thin on content, not nearly addressing the growing body of literature on selectively driven pop divergence, depth associated pop divergence, and even failing to describe the results of a very relevant study by Sivasundar & Palumbi in any detail. The methods are mostly a list of references. Most of the results are relegated to supplementary material, with only 1 figure and 1 table in the manuscript itself. Finally, the conclusions are tenuous and would benefit from additional analyses – such as analysis of other loci already published, and implementing additional methodologies.
I would be happy to review and provide constructive criticism on a more fleshed out version of this work.

Experimental design

2 samples of 25 individuals is not ideal, but the results are compelling.

Validity of the findings

I'm fine with the rhodopsin findings. My major sticking point is the interpretation of what is happening at the population level based upon mtDNA and the loci investigated by previous researchers

Additional comments

Abstract
Overall, the abstract needs work on the wording.
Third sentence is awk. Do you mean that a population is partitioned by depth with 550 meters being the depth where partitioning occurs?

Introduction
The introduction is brief and to the point, but also suffers from some glossing over of the concepts being addressed and the cited literature is thin. The explanations of concepts like sympatric, parapatric, and allopatric divergence should be revised to be more clear.
8: diverging taxa
8-10: There is additional literature on this subject, such as Bird et al. 2011. Molecular Ecology 20., Bird et al 2012 Evolutionary Biology. 39 is a review and should provide additional references
11: Populations diverging in sympatry. There are several sympatric populations in any location.
13: awk. Parapatric divergence is not a complex process, please explain more clearly
15-17: exception is Bird et al 2011, as well as Bird 2011 Journal or Integrative and Comparative Biology. 51. See lit cited in Bird et al 2012 mentioned above http://link.springer.com/article/10.1007/s11692-012-9183-6
18: What are adaptive challenges? Does this mean selective pressure?

Methods
I’m a strong supporter of brevity, but two paragraphs on materials and methods for a pop gen project involving two sequenced loci addressing? The reader should not have to track down several other publications to know what was done in the present manuscript.

IMa2 would be a perfect model simulation to analyze this data set and employs the coalescent, which is not implemented in present analyses.

Results
64: Is it haplotypes or alleles?
65: there is specific terminology for describing a “star-like structure”. Starburst. It’s typically indicative of an expanding population
68-69: please redefine SP and DS in each section of manuscript
71: FST? I didn’t see that in the methods. What method of FST was employed? Did you take sequence differences into account? Or did you treat each haplotype as equally different? If the former, did you implement a substitution model?
77-78: I’d like to see how IMa2 results compare
79-80: There’s only one figure here. Why does this figure need to be supplemental? Same thing with the tables. No need to make them supplemental as they are key to my understanding of the results as the reader. So now, not only do I have to dig up other publications to figure out what was done (see methods) but now (except for figure 1) I have to go and find the supplemental materials to see the results.
114-115: Perhaps the mitochondrion is experiencing disruptive selection? It seems that neutrality of mtDNA is being assumed, but if other loci didn’t detect the depth partitioning then there may be no neutral divergence/ genetic drift.
131: awk




Fig 1 is not labeled properly. Each figure within the figure should be labeled. A, B,C D. It is not 100% clear which locus is represented in the graphs. Also, the depth axis is likely not reflective of where the haplotypes and alleles were found in the network – thus it is misleading. Instead, there should be a simple color key.

Reviewer 2 ·

Basic reporting

I am unclear why a supplementary materials section was used. Includ this in the manuscript. Figure S1 needs to be in the manuscript as well.

All figures and tables need to stand on their own. Please provide more detail where appropriate.

Line 118 - you do not discuss migrations of these fish at different stages of the life cycle. This should be included in the introduction. Do you have any data on where in the life cycle these fish were and whether this stage indeed migrates between depths? If not, how do you come to this conclusion?

Minor comments

No mention of the Genbank ascension numbers anywhere in the text. Please include.

1. line 1. change "Incipient Speciation" to "Isolation". Remove the "?"
2. line 17. You may also find Brokovich et al. 2010. Phys. and Behav. 101:413- 421 of interest.
3. line 18 - replace "along the water column" with "between depths"
4. line 25 - replace "has" with "have"
5. line 26 - replace "assisted" with "associated"
6. line 27 - period after et al
7. line 27 - replace "]" with ")"
8. line 32 - period after et al
9. line 33 - replace "]" with ")"
10. line 45 - comma after "et al."
11. line 50 - For which region? Specify.
12. line 52 - add space after "ARLEQUIN"
13. line 54 - comma after "von Haeseler A."
14. line 55 - change "investigated by means of" to "conducted using"
15. line 56 - delete "the program"
16. line 60 - space after "2010)"
17. line 61 - add "the" before "statistical"
18. line 76 - add actual % allele divergence
19. line 79 - need better clarification of transition into Rhodopsin section.
20. line 80 - change "represents" the closest to "shares the most recent"
21. line 97 - remove the "a" from "in a shorter" and "suggesting a more"
22. line 114 - add a period after "et al"
23. line 118 -
24. line 123 - State what was previously hypothesized.
25. line 124 - replace "]" with ")"
26. line 128 - Clarify. How did increasing fishery pressure on the stocks disregard substructure?
27. line 131 - change "lineages ongoing a process" to "lineages in an ongoing process"
28. line 216 -should be p< ?


Supplemental information

1. line 11 - how were these sub-sampled
2. line 13 - add "using a" after "EtOH"
3. line 25 - add "]" after "(45 sec)"
4. line 37 - Please be clear if the table refers to rhodopsin or the control region. Verify that all figures and tables are similarly defined.
5. line 44 - Remove "in the study"
6. line 54-57 - Starting with "We used" awkward wording please rephrase.
7. line 61 - See comment 4 above; reduce number of sig. digits in p-values
8. line 67 - Replace "the best fit model the data on ModelTest 3.7" with "using Modeltest3.7, the model that best fit the data was"
9. line 69 - Remove "under the"
10. line 73 - replace "was tested using Creevey" to "were tested using the Creevey"
11. line 76 - add "of" after number
12. line 83 - remove ( ) from around "Fisher's exact test"
13. line 88 - remove "rooted"
14. line 89 - alutus should not be capitalized
15. line 107 - Sebastes should be italicized
16. line 114 - "Rhodopsin" should be rhodopsin here and elsewhere throughout the text.
17. line 115 - the asterisks are not in the figure

Experimental design

Figure 1S shows that the two out-group species are actually paraphyletic with the deep S. mentella. At what depth were individuals of the outgroup species collected? According to fishbase S. viviparous depth only extends to 300m, well within the "shallow" populations range. You need to explain this result. Would this still be considered incipient speciation if this substitution is prevalent throughout the genus with or without a depth related divergence? Please discuss.

How were samples "randomly selected"? Were they all from the same location or multiple fishing grounds? How do you know that the variation you see isn't due to location rather than depth?

Validity of the findings

The authors have two goals. First, they aim to reconstruct historical demography between deep and shallow fishes, which they have done with mtDNA. Second, they aim to investigate whether disruptive selection has acted on the rhodopsin gene in this species causing incipient speciation. If the authors achieved their aim, this would be a very interesting result. However, the conclusions the authors have come to are premature. This manuscript would benefit from refocusing on what was actually tested and leave incipient speciation and disruptive selection for a discussion point and future direction. The authors have strong evidence to show isolation between shallow and deep individuals. Focus on this in the paper.

Additional comments

This paper aims to shed light on the mechanisms responsible for the genetic partitioning observed in previous work between "deep" and "shallow" individuals of Sebastes mentella. As the authors state, little attention has been given to the role depth plays in marine systems, making this a novel and potentially important study. This study is very interesting and with a little work I believe it will be a valuable contribution.

Reviewer 3 ·

Basic reporting

no comments

Experimental design

no comments

Validity of the findings

no comments

Additional comments

The manuscript presented by Shum and colleagues deals with the genetics of shallow and deep populations of an Atlantic Rockfish. A mitochondrial marker (Dloop) and a nuclear gene (Rhodopsin, a visual pigment gene) were used on a relatively small sample size (although it is likely that deep water samples are not very simple to obtain.
Overall the paper is tantalizing and deserves publication. Results, however, need to be discussed more cautiously in my mind.
Population structure is very strong. Even with relatively small sample sizes, the two populations partition very strongly, as shows by the mitochondrial marker. In fact the two markers show essentially the same networks. This is the hallmark of non interbreeding populations, where reciprocal monophyly is expected at most markers. Some deep individuals with shallow haplotypes were described as potential vagrants, reinforcing the idea that full sorting has occurred. If this were the case, the argument in favor of positive selection on the Rhodopsin becomes weaker. It is likely that another nuclear marker would also show fixed differences, although not associated with vision. The fact that the fixed nucleotide happens to change an amino acid is very interesting and suggestive of selection, obviously. yet its non-essential position in the protein indicates that its role may not be very important and may simply be a case of random sorting.
So overall I find this paper interesting and worthy of publication. However, the small sample size and the use of very few molecular markers precludes a strong conclusion. Instead, a cautious prediction of the potential causes for these results will serve as a nice introduction to further studies.

External reviews were received for this submission. These reviews were used by the Editor when they made their decision, and can be downloaded below.

---

## Round 0.2 · Minor Revisions

· Academic Editor

Minor Revisions

Both referees agreed that the revisions have greatly improved the manuscript and the paper should now be acceptable pending some last minor revisions that each felt was important enough to recommend an additional round of minor revisions rather than direct acceptance at this point. However, neither felt the need to see the manuscript again after the authors deal with these last issues. I agree with both referees on the issues that they raise as important here (particularly the number of permutations in your analyses and appropriate caveats in the discussion), but I expect that it should not take long to turn this around, and I do not intend to send it back to the referees at this point. I look forward to seeing your revised manuscript so that I can move it forward into production.

Reviewer 1 ·

Basic reporting

pass

Experimental design

pass

Validity of the findings

pass

Additional comments

Thank you for taking my first round of comments to heart and editing the manuscript accordingly. I think that this is a fine contribution to the growing evidence that population partitioning along depth gradients is important. Minor comments follow.

19-21 – it should be noted that some or all of these manuscripts look at the role of depth gradients. As cited right now, they might only say that depth gradients have received little attention, which is a little misleading.
74 – 1000 permutations is not very much. This means the lowest p value possible is 0.001 (or <0.001)
86- F in FST should be italicized. D in Dest should be italicized. Make sure this is consistent throughout. The equation editor in Microsoft word can help to check your formatting of equations, parameters, and variables. Because multiple stats are calculated with the same data, there should be some sort of false discovery rate correction. Based upon a cursory look at your pvalues, Bonferroni correction probably won’t change your results, but the critical p value will be close to 0.001, the minimum level of resolution with 1000 permutations. If Bonferroni is too extreme, Benjamini has several more recent manuscripts describing modern FDR correction.
87- typo in GenALEX, again 999 permutations is low.

186: This is personal preference, but I think this sentence would read better like this:
These latest results open the possibility that the ‘shallow’ and ‘deep’ groups may represent diverging lineages experiencing adaptation towards divergent environmental conditions.
Figure 1: italicize species name and spell out genus name when it is first word in sentence
Figure 3. italicize species name
Table 1. The order of the columns doesn’t match the order in which they are described. There a ? in the column labels in the table
Table 2. italicize species name, check throughout. calculate Dest for mtDNA. Why not calculate these stats for the rhodopsin locus?

Reviewer 2 ·

Basic reporting

This revision of the manuscript is greatly improved from the previous draft.


One minor comment

Line 93: the citation (Posada and Crandall, 1998) should follow directly after Modeltest 3.7

Experimental design

One of the weakness of this paper is in the small sample sizes (10 shallow and 12 deep) for the rhodopsin gene especially given the fact that the samples were readily available. Given the authors choice to forego the recommendation to increase these numbers, and the fact that they were unable to conduct the more rigorous analyses recommended by the reviews due to the limitations of their sample sizes, it should be clearly stated in the manuscript that the authors acknowledge the possible issues of small sample sizes and their inability to conduct the recommended analyses because of it.

Validity of the findings

The authors state in their revision letter that "THE LIFE CYCLE OF OCEANIC S. MENTELLA STILL REMAINS TO BE FULLY UNRAVELLED. WE HAVE NO SOLID DATA ON THIS, AND SPECULATING ON WHAT DEEP AND SHALLOW UNIT MIGHT DO, IN TERMS OF MATING AGGREGATIONS, LARVAL RELEASE, ETC. IS BEYOND THE REACH OF THIS INVESTIGATION." However, in the manuscript (line 169) they continue to do just that by stating "which can be interpreted as the occurrence of individual movements along the water column during the life cycle (i.e. short-term "dives" into the deep, by shallow-dwelling fish). I agree with their assessment that this statement is beyond the reach of this investigation and again encourage them to remove/rephrase this statement.

---

## Round 0.3 · accepted · Accept

· Academic Editor

Accept

Thank you for responding the the referee comments. I am satisfied that you have taken due diligence in your revisions, and feel that your manuscript will be a valuable contribution to the literature. I am happy to accept your paper and move it forward for publication.